**∂ | Open Peer Review | Biotechnology | Research Article**

# Ultrasensitive one-pot detection of monkeypox virus with RPA and CRISPR in a sucrose-aided multiphase aqueous system

Yue Wang,[1] Yixin Tang,[1] Yukang Chen,[1] Guangxi Yu,[1] Xue Zhang,[1] Lihong Yang,[1] Chenjie Zhao,[1] Pei Wang,[2] Song Gao[1]

**ABSTRACT**  The monkeypox virus, which was declared as a Public Health Emergency of International Concern (PHEIC) by the World Health Organization (WHO), continues to cause infection cases worldwide. Given the risk of virus evolution, it is essential to identify monkeypox virus infection in a timely manner and isolate patients to prevent outbreaks so that the vulnerable population is protected and the emergence of dangerous monkeypox variants is restrained. In this study, a novel one-pot recombinase polymerase amplification-Clustered Regularly Interspaced Short Palindromic Repeats (RPA-CRISPR) assay for monkeypox based on a sucrose-aided multiphase aqueous system has been established with an ultra-high sensitivity of a single copy, which is 10 times higher than the existing RPA-CRISPR one-pot method. The entire reaction was completed within 60 min at 37°C. The detection method exhibited good specificity, effectively distinguishing monkeypox from other orthopoxviruses. The detection results could be observed by the naked eye under ultraviolet or blue light, making it highly suitable for home or limited healthcare settings. The assay has solved the incompatibility between the Cas12a cleavage reaction and the RPA reaction and shows good specificity, accuracy, and the rapidness and convenience important for point-of-care testing. It provides an effective tool for the early diagnosis of monkeypox and sets a technical example of using a multiphase aqueous system to establish one-pot RPA-CRISPR detection.

**IMPORTANCE**  The monkeypox virus was declared as a Public Health Emergency of International Concern (PHEIC) by the World Health Organization (WHO) and continues to cause infection cases worldwide. Given the risk of virus evolution, it is essential to identify monkeypox virus infection in a timely manner to prevent outbreaks. This study establishes a novel one-pot recombinase polymerase amplification-Clustered Regularly Interspaced Short Palindromic Repeats (RPA-CRISPR) assay for monkeypox virus with an ultra-high sensitivity. The assay shows good specificity, accuracy, and the rapidness and convenience important for point-of-care testing. It provides an effective tool for the early diagnosis of monkeypox, which is useful for the prevention of an epidemic.

**KEYWORDS**  monkeypox, one-pot detection, RPA-CRISPR, multiphase aqueous system, POCT

T he monkeypox virus was declared as a Public Health Emergency of International Concern (PHEIC) by the World Health Organization (WHO) and continues to cause infection cases worldwide, posing a major public health challenge (1). By October 2023, over 90 thousand infections have been reported to the WHO from 115 countries and territories, including hundreds of deaths (https://worldhealthorg.shinyapps.io/mpx_global). The virus causes monkeypox, a zoonotic infectious disease with smallpox-like symptoms, including rash, fever, headache, and muscle pain (2–5). In most cases, the symptoms are mild, but rare life-threatening complications, such as pneumonia, central nervous system infection, and organ failure, can still occur (6, 7). The

Address correspondence to Pei Wang, 90830@njnu.edu.cn, or Song Gao, gaos@jou.edu.cn.

Yue Wang and Yixin Tang contributed equally to this article. Author order was determined by drawing straws.

The authors declare no conflict of interest.

See the funding table on p. 14.

transmission of monkeypox through direct contact and respiratory droplets can easily lead to an epidemic (8–10). Given the risk of virus evolution, it is essential to identify monkeypox virus infection in a timely manner and isolate patients to prevent outbreaks so that the vulnerable population is protected and the emergence of dangerous monkeypox variants is restrained.

In this regard, early diagnostic measures are important for the prevention of the spread of monkeypox and the protection of public safety. Nucleic acid detection has become a fundamental tool for the early diagnosis of monkeypox because of its rapid response and diagnostic certainty; similar viruses, such as smallpox, can be effectively distinguished (11, 12). However, the current nucleic acid detection assays for monkeypox are not satisfactory enough. The PCR assays are reliable but are laboratory-based because precise thermocyclers are required (13, 14). The isothermal amplification assays, such as loop-mediated isothermal amplification (LAMP) and recombinase polymerase amplification (RPA), are vulnerable to amplification interference and thus are not sufficiently reliable for point-of-care testing (POCT) (15–17).

The Clustered Regularly Interspaced Short Palindromic Repeats (CRISPR)/Cas12a system has been widely applied to the early diagnosis of pathogen infections (18–20). The system relies on the recognition of a pathogen-specific sequence by the Cas12a-crRNA complex followed by the activation of Cas12a to cleave a signal-producing probe. This system is highly specific because the activation of Cas12a depends on a good match between the rationally designed crRNA and the target sequence from the pathogen. Thus, by combining the CRISPR/Cas12a system with isothermal amplification, the detection accuracy and reliability can be greatly improved. Famous application examples of this kind of combination include the detection of African swine fever virus and SARS-CoV-2 (21, 22).

A highly sensitive diagnostic assay for monkeypox that combines RPA with CRISPR/Cas12a has been reported (23). The sensitivity has achieved the single-copy level, but the assay uses a two-step procedure, which not only is operationally complicated but also has the risk of carry-over contamination. The further developed RPA-CRISPR one-pot assay has sacrificed the sensitivity because of an incompatibility between amplification and Cas cleavage, which has decreased the sensitivity to a dozen-copies level (24). In this study, a novel RPA-CRISPR one-pot assay for monkeypox based on a sucrose-aided multiphase aqueous system has been established with an ultra-high sensitivity of a single copy. The assay has solved the incompatibility problem and shows good specificity, accuracy, and the rapidness and convenience important for POCT. It provides an effective tool for the early diagnosis of monkeypox and sets a technical example of using a multiphase aqueous system to establish one-pot RPA-CRISPR detection.

## RESULTS AND DISCUSSION

### Principle of the sucrose-aided multiphase one-pot RPA-CRISPR reaction

The combination of RPA and CRISPR technology in the development of a one-pot detection method often leads to a loss of sensitivity, which may be due to the incompatibility between Cas12a cleavage and RPA (25, 26). In the early stage of the one-pot reaction, Cas12a can form a complex with specific targets under the guidance of crRNA, thereby activating the *cis* and *trans* cleavage activities of Cas12a. The specific *cis* cleavage activity leads to the loss of RPA template DNA and the difficulty in accumulating newly generated RPA amplicons. The non-specific *trans* cleavage activity leads to the cleavage of primers in the reaction system, resulting in reduced RPA efficiency. For example, Mao et al. reported a two-step method for detecting monkeypox using RPA combined with Cas12a, with a sensitivity of $10^0$ copies per reaction (23). In contrast, Chen et al. established a one-step method for detecting monkeypox using RPA combined with Cas12a, with a sensitivity of 15 copies per reaction (24).

In this study, to address the challenge of overcoming the incompatibility between the RPA reaction and the CRISPR reaction, we established a sucrose-aided multiphase aqueous system for the one-pot detection of the monkeypox virus (Fig. 1). By taking

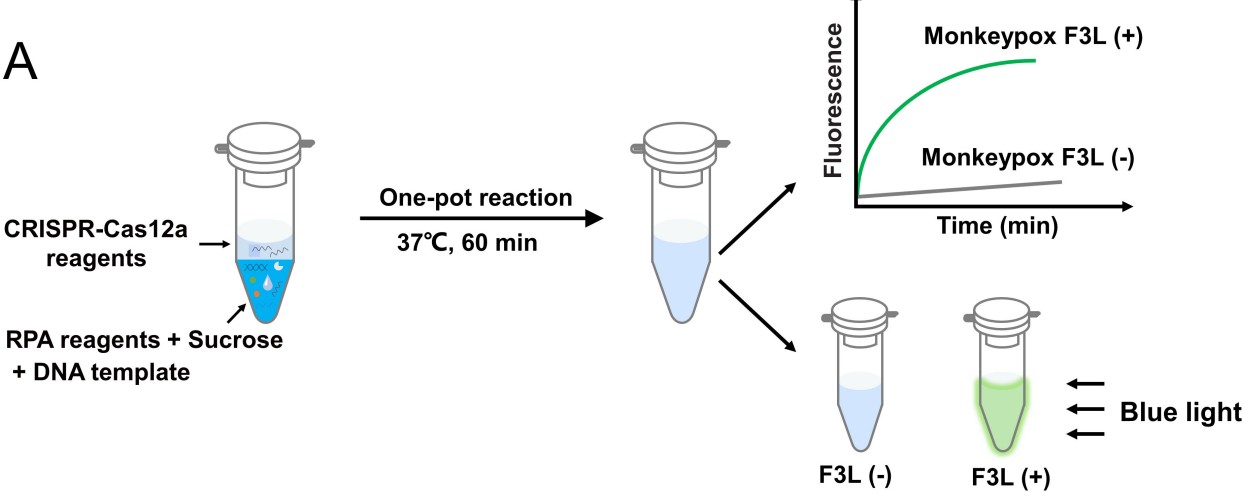

**FIG 1** Schematic of the RPA-CRISPR one-pot reaction with the sucrose-aided multiphase. (A) The reaction flowchart. The RPA and CRISPR reaction components in the two phases of the same tube are shown in different colors. The real-time and end-point fluorescence detections are depicted. (B) The diagram showing the RPA and CRISPR reactions separated by the dynamic interface of the two phases. The components in the reaction are represented by different shapes. Base complementary pairing between the crRNA and the target is shown. The specific Cas12a cleavage sites on the target are represented by red scissors.

advantage of the density difference between sucrose and water, we were able to spatially separate the RPA reaction and the CRISPR reaction within the same tube. The RPA reaction, with the addition of sucrose to increase its density, was placed at the

bottom of the tube, while the CRISPR reaction, with lower density, remained in the water phase and was positioned on top of the RPA reaction. During the one-pot reaction, slow and dynamic spreading occurred between the two phases. The newly accumulated RPA amplicons triggered the activation of the non-specific *trans* cleavage of Cas12a, resulting in the cleavage of ssDNA-FQ and the release of the fluorophore; meanwhile, the phase separation diminished the effect on RPA. Real-time fluorescence signals were detected using a fluorescence detector, while the end-point fluorescence could be visualized under a blue light source.

Various methods based on the principle of multiphase aqueous systems have been reported to overcome the compatibility issue between RPA and CRISPR cleavage (27, 28). Yin et al. reported a molecular diagnostic method for CRISPR-Cas12a using a multiphase aqueous system, dynamic aqueous multiphase reaction (DAMR), which utilizes the density difference based on sucrose concentration (28). In the DAMR system, RPA- and CRISPR-Cas12a-derived fluorescence detections occur in spatially separated but connected aqueous phases. This method detected 10 copies of HPV16 and 100 copies of HPV18 in less than 1 h (28). Lin et al. reported a one-pot RPA-CRISPR method developed using a multiphase aqueous system based on the density difference associated with glycerol concentration. This method demonstrated a 100-fold improvement in sensitivity compared to the glycerol-free approach (27). These examples demonstrated the applicability of density-based multiphase systems for solving the RPA-CRISPR compatibility issue. Sucrose and glycerol are traditional enzyme protectors, and studies have shown their enhancing effect on nucleic acid detection (29, 30). Our study further proved that sucrose would not interfere with CRISPR cleavage.

## Screening of the RPA primers

The three forward primers (F1–F3) and three reverse primers (R1–R3) (Table 1) could make nine primer pair combinations, which were tested in PCRs. The results of the 1.5% agarose gel showed that the amplification efficiency of the reactions using the combinations of F1R1, F1R2, and F3R2 was higher than that of the other combinations (Fig. 2A). Subsequently, the three primer pairs were individually re-confirmed in the RPA reactions, and the results of the 1.5% agarose gel showed that the amplification efficiency of the RPA reaction with the primer pair F1R2 was the highest (Fig. 2B). Finally, the sensitivity of the RPA reaction with the primer pair F1R2 was tested, and the results showed that the amplification efficiency was good, with as low as $10^1$ copies of the template (Fig. 2C). Therefore, the primer pair F1R2 was selected for the subsequent experiments.

## Screening of the crRNA

After determining the RPA primer, the DNA sequence of the RPA amplicon was analyzed, and four crRNAs were designed (Table 1). To establish a highly efficient CRISPR reaction,

**TABLE 1** Oligonucleotide sequences[a]

| Reaction | Name | Sequence (5'–3') | Length (nt) |
|---|---|---|---|
| RPA | MPXV-F1 | CGAGAAGTTAATAAAGCTCTGTATGATCTTCAACG | 35 |
| | MPXV-F2 | CTCTGTATGATCTTCAACGTAGTGCTATGG | 30 |
| | MPXV-F3 | CTTCAACGTAGTGCTATGGTTTACAGC | 27 |
| | MPXV-R1 | CGTTTAGATTTTCCATCTGCCTTATCGAATACTC | 34 |
| | MPXV-R2 | CCTTATCGAATACTCTTCCGTCAATGTC | 28 |
| | MPXV-R3 | CATCTCGTTTAGATTTTCCATCTGCCTTATC | 31 |
| CRISPR/Cas12a | crRNA1 | *UAAUUUCUACUAAGUGUAGAU*CAGCUCCAACGAUACUCCUCC | 42 |
| | crRNA2 | *UAAUUUCUACUAAGUGUAGAU*AUGAUGUUAUUCCGGUUAAAA | 42 |
| | crRNA3 | *UAAUUUCUACUAAGUGUAGAU*UUGGAAAGGUGUUAACCCUGU | 42 |
| | crRNA4 | *UAAUUUCUACUAAGUGUAGAU*GUAUUGAAUCAGUGGGGCCUA | 42 |
| | ssDNA-FQ | FAM-TTATT-BHQ1 | 5 |

[a]The underlined sequence is the crRNA skeleton.

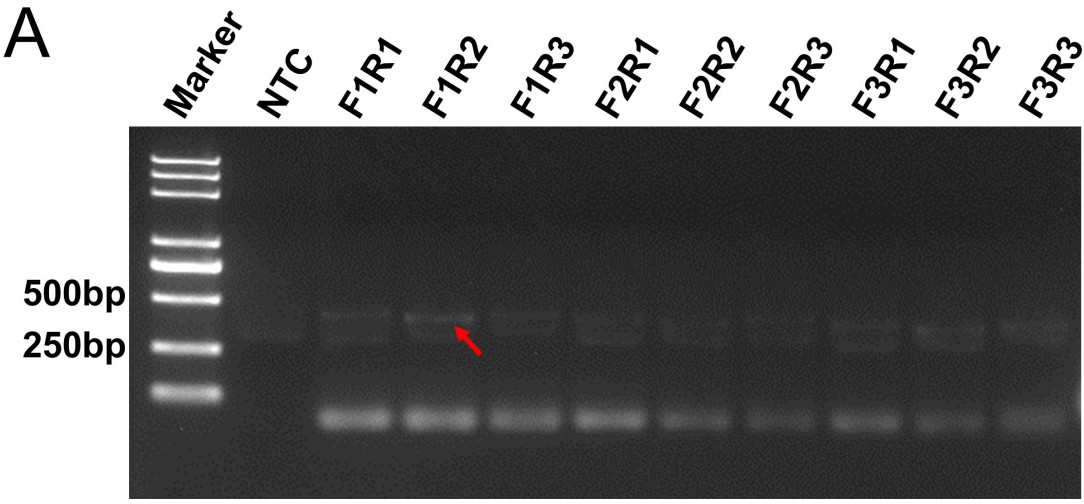

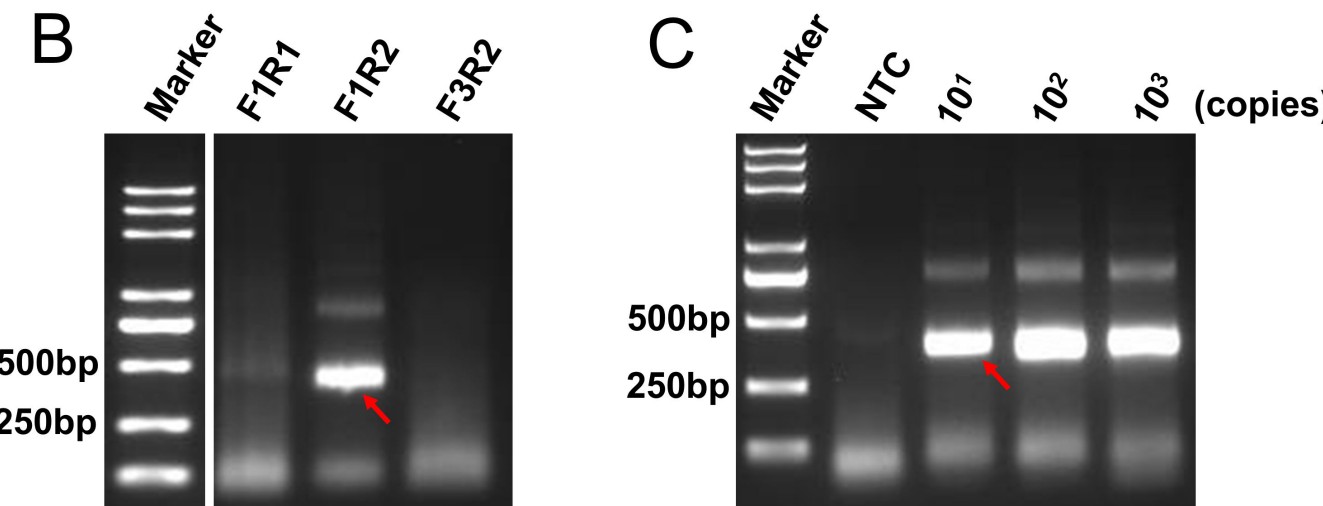

**FIG 2** RPA primer screening. (A and B) Primer screening with PCR (A) and RPA (B) Agarose gel image showing the amplification efficiency of different primer combinations, with the names of the primer pairs used in the reaction displayed above the respective lanes. The template amount used was $10^3$ copies per reaction. (C) Amplification efficiency of primer pair F1R2 in the RPA reaction. The standard plasmid template amounts used in the reactions are displayed above the respective lanes. Amplification bands corresponding to the specific amplicons are indicated by red arrows. NTC stands for the no-template control.

the four crRNAs were tested individually. The results of the fluorescence curves showed that with the guidance of crRNA1, the cleavage activity of Cas12a was the highest, and the end-point fluorescence could be visualized under a blue light source (Fig. 3A). Then, the sensitivity of the two-step RPA-CRISPR method was determined by using different concentrations of the standard plasmid. Both the results of the fluorescence curves and the end-point fluorescence showed that as few as $10^1$ copies of the standard plasmid could be detected (Fig. 3B).

## Construction of the one-pot RPA-CRISPR reaction in a sucrose-aided multi-phase aqueous system

Based on the two-step RPA-CRISPR reaction, efforts were made to establish a one-pot RPA-CRISPR reaction. First, $10^5$ copies of the standard plasmid were used as the templates and tested in the one-pot RPA-CRISPR reaction that directly combined the RPA reaction

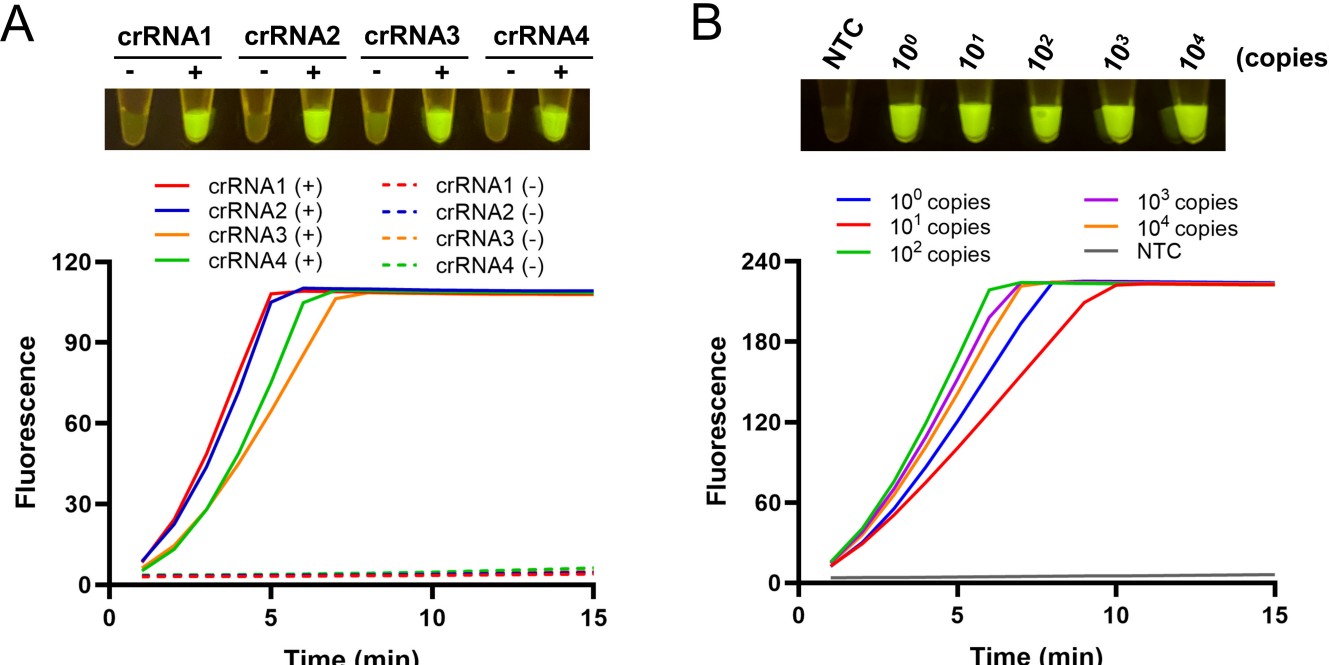

**FIG 3** crRNA screening and confirmation. (A) Screening of crRNAs 1–4. Both the fluorescence curves and the end-point fluorescence signals are shown. The template used was $10^4$ copies per reaction of the standard plasmid. (+) represents the addition of the template in the reactions, and (-) represents the no-template controls. (B) Confirmation of crRNA1 in the sensitivity test. The amounts (in copies) of the standard plasmid template used for the reactions are indicated. NTC represents the no-template control.

and the CRISPR reaction. However, the results showed no obvious difference between the fluorescence curves of the positive reaction and the no-template control (Fig. 4A). As we have speculated, the reason could be that in the one-pot mixture containing the RPA reagents and the CRISPR reagents, the newly produced RPA amplicons would be cleaved by Cas12a, resulting in poor accumulation of the amplicons and a low fluorescence signal (25, 26).

To solve this problem, sucrose was introduced to construct the multiphase aqueous system. The sucrose density is $1.8 \pm 0.1$ g/cm³, which is higher than that of water. The density of the RPA reaction with sucrose was higher than that of the CRISPR reaction in the aqueous phase. Additionally, sucrose demonstrated stability in both physical and chemical properties. Here, the sucrose-aided multiphase aqueous system was constructed with the CRISPR reaction in the upper phase and the RPA reaction in the bottom phase and was confirmed by using a template of $10^4$ copies of a standard plasmid (Fig. 4B). Indeed, the good fluorescence signal was related to the better RPA in the multiphase aqueous system. The multiphase reaction (+ sucrose) that showed a good fluorescence signal did have much better accumulation of the RPA amplicons as compared to the monophase reaction (- sucrose) (Fig. 4C and D).

After the construction of the multiphase aqueous system, a series of optimization experiments was conducted. These experiments included optimizing the concentrations of sucrose and the ratio of the RPA reaction to the CRISPR reaction. The optimization of sucrose concentrations was tested by adding different concentrations of sucrose into the RPA reactions. The results showed that the one-pot reaction with 5% sucrose in the RPA (sucrose) phase exhibited the highest fluorescence signals (Fig. 5A). For the optimization of the ratio of the RPA reaction to the CRISPR reaction, three different ratios were tested: 1.5:1, 2:1, and 3:1. The results indicated that the reaction with a ratio of 2:1 for the RPA reaction to the CRISPR reaction displayed the highest fluorescence signals (Fig. 5B). With regard to the concentrations of Cas12a and crRNA, the literature and the instructions for commercialized Cas12a (New England Biolabs) have recommended a molar ratio of ~1:1

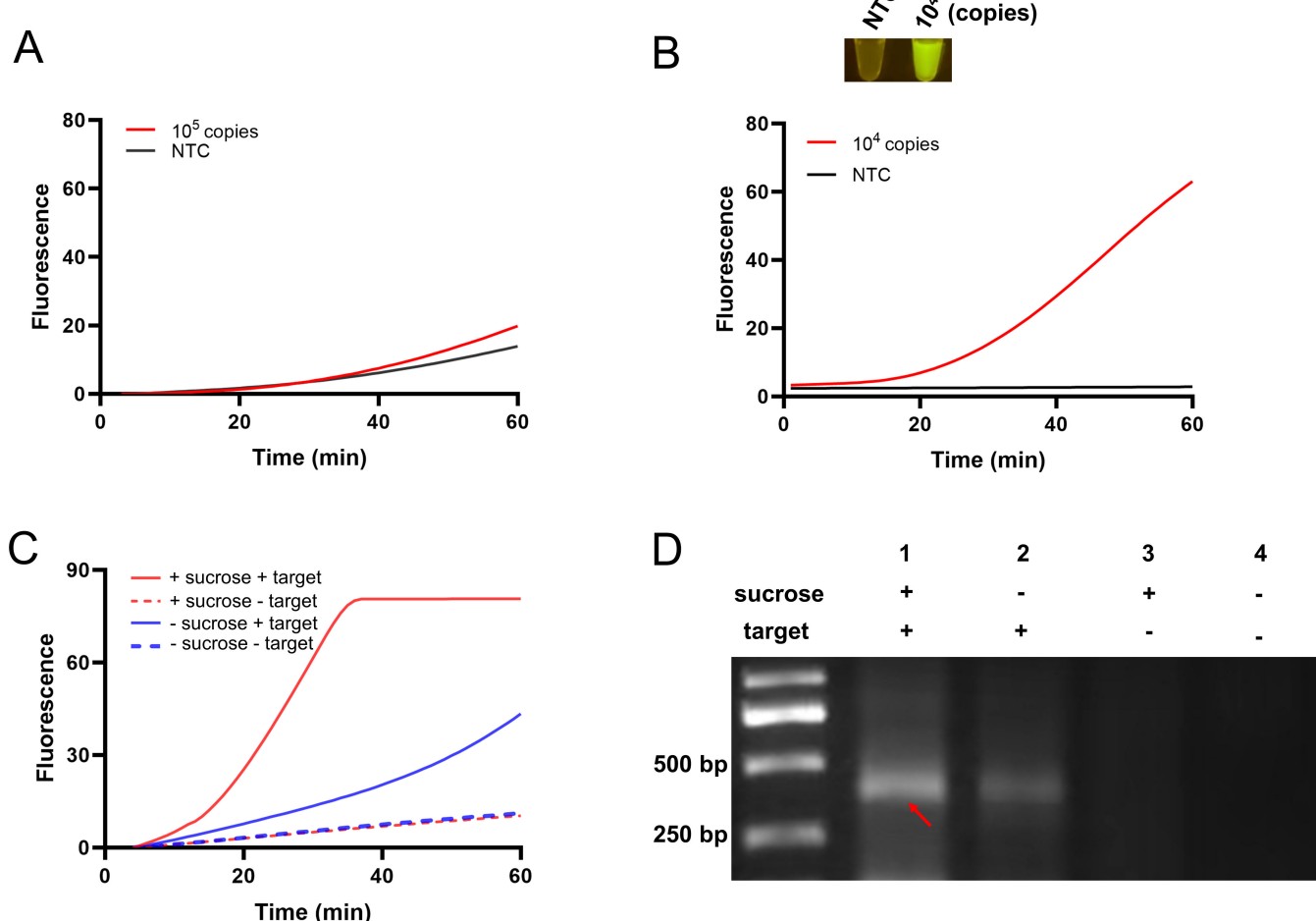

**FIG 4** Construction of the one-pot RPA-CRISPR reaction in a sucrose-aided multiphase aqueous system. (A) Direct combination of the RPA and CRISPR reactions into one pot. The fluorescence curves are shown. The standard plasmid template amount ($10^5$ copies) used is indicated. (B) One-pot reaction with the sucrose-aided multiphase. Both the fluorescence curves and the end-point fluorescence signals are shown. The template amount is indicated. NTC represents the no-template control. (C) Validation of the one-pot reaction with the sucrose-aided multiphase. The fluorescence curves with (+ sucrose) or without (- sucrose) the sucrose-aided multiphase are shown. The reactions with (+ target) or without (- target) the template are indicated. (D) Agarose gel image showing the RPA amplicons from the reactions in panel C.

in the range of 30–50 nM final (20, 24, 31). This study selected the molar ratio of 1:1 at 33 nM final for Cas12a and crRNA concentrations.

## Sensitivity and specificity of the sucrose-aided one-pot RPA-CRISPR method

To evaluate the limit of detection of the sucrose-aided multiphase one-pot RPA-CRISPR method, a series of different copies of the standard plasmid of the monkeypox virus, including $10^5$, $10^4$, $10^2$, $10^1$, and $10^0$, were tested. The results of the fluorescence curves and the end-point fluorescence showed that as few as $10^0$ copies of the standard plasmid could be detected (Fig. 6A and B). The significance analysis further indicated that the positive signals were solid.

The specificity of the sucrose-aided one-pot RPA-CRISPR method was confirmed by testing a series of commonly encountered vaccinia viruses and coronaviruses, including cowpox virus, buffalopox virus, vaccinia virus, variola virus, SARS-CoV-2 N gene, SARS-CoV-2 ORF gene, and *Pseudomonas aeruginosa*. Both the results of the fluorescence curves and the end-point fluorescence demonstrated that this method was specific to the monkeypox virus (Fig. 6C and D).

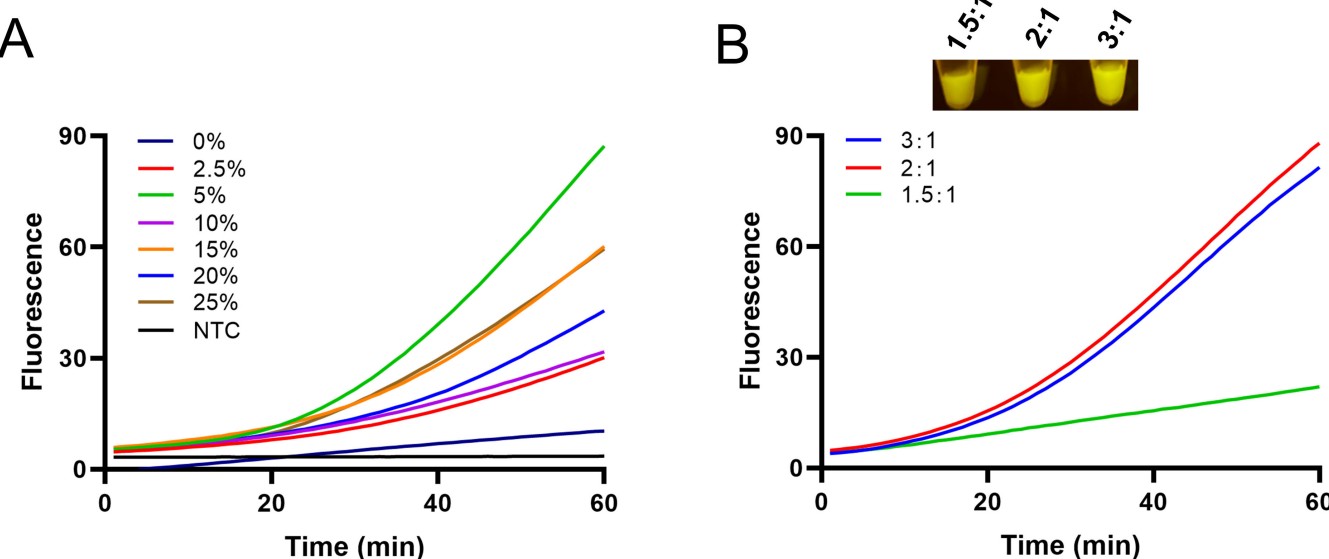

**FIG 5** Optimization of the one-pot RPA-CRISPR reaction with the sucrose-aided multiphase. (A) Optimization of the sucrose concentration in the RPA (sucrose) phase. The fluorescence curves are shown for reactions with different concentrations of sucrose. Template amount: $10^4$ copies of the standard plasmid per reaction. NTC represents the no-template control. (B) Optimization of the volume ratio of the RPA (sucrose) phase and the RPA-CRISPR phase. The fluorescence curves are shown for reactions with different ratios.

## Validation of the sucrose-aided one-pot RPA-CRISPR method with simulated samples

Since the monkeypox virus is presumed to spread through close contact with bodily fluids, we prepared three simulated samples by mixing the bodily fluids from blood samples, skin swabs, and throat swabs with the monkeypox standard plasmid and one simulated sample by mixing the skin swab with the monkeypox pseudovirus. These samples were then tested to validate the sucrose-aided one-pot RPA-CRISPR method. The fluorescence curves and the end-point fluorescence results indicated that all four types of simulated samples could be detected with single-copy sensitivity, demonstrating the potential of the one-pot method for clinical diagnosis of the monkeypox virus (Fig. 7).

## Advances in technology and application

CRISPR-Cas12a has shown good performance in nucleic acid detection when combined with nucleic acid amplification technologies (32–35). The specific activation of Cas12a cleavage guided by crRNA and the signal amplification effect are the main improving factors. To make this amplification-cleavage detection system more convenient and to reduce the risk of carry-over contamination, one-pot assays are the technology development direction. Specifically, the development of RPA-CRISPR one-pot detection methods requires overcoming the compatibility issue between the RPA reaction and the CRISPR cleavage reaction. Lin et al.'s study implemented a single-tube system for the RPA and CRISPR reactions in a glycerol-based double aqueous system, improving the reaction sensitivity by 100 times (27). Yin et al. developed a double aqueous system utilizing sucrose density to overcome the incompatibility between the RPA and CRISPR reactions, enabling highly sensitive and quantitative detection (28). Gong et al. used sulfur-modified RPA primers to avoid detection failures caused by non-specific cleavage of RPA primers by Cas12a (36). Lu et al. established a rapid and highly sensitive one-pot method for the RPA and CRISPR reactions based on suboptimal protospacer adjacent motif (PAM) sequences (37). Chen et al. reported a method that temporarily inactivates crRNA by introducing photocleavable nucleotides complementary to crRNA (25). These attempts

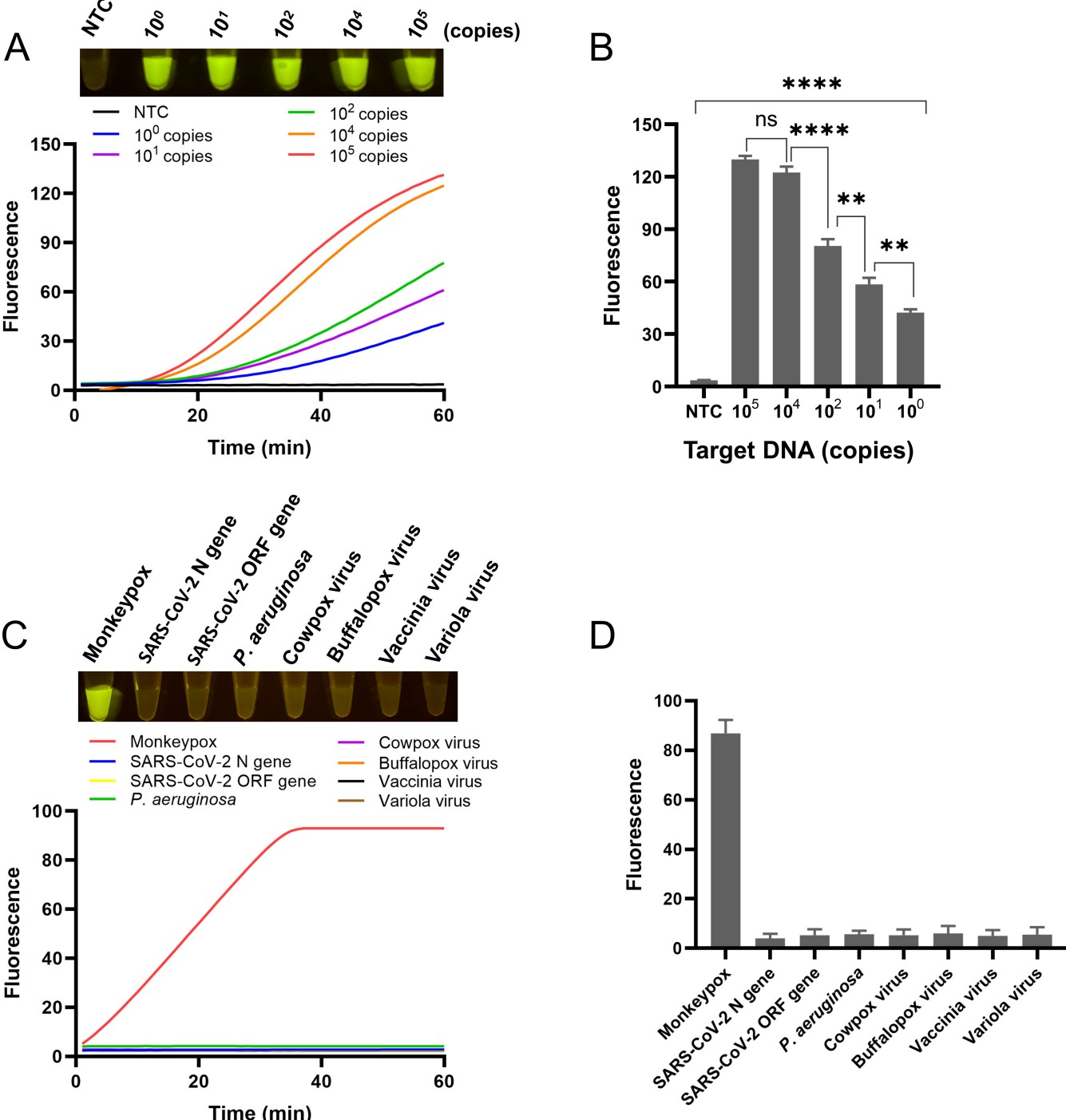

**FIG 6** Sensitivity and specificity. (A) Sensitivity test. Fluorescence curves and end-point fluorescence signals of reactions with different standard plasmid template amounts are shown. (B) Bar chart of the end-point fluorescence intensities from the sensitivity test. Error bars represent the standard error of two replicates. ****$P < 0.001$, **$P < 0.01$, and $^{ns}P = 0.2$. (C) Specificity test. Fluorescence curves and end-point fluorescence signals of reactions with different templates are shown. (D) Bar chart of the end-point fluorescence intensities from the specificity test. Error bars represent the standard error of two replicates.

to combine the RPA and CRISPR reactions in a one-pot method provide successful cases for the application of CRISPR technology in on-site detection scenarios. In this study, we constructed the multiphase aqueous system using sucrose, enabling the stratification of the RPA reaction system and the CRISPR reaction system in a single tube with a gradual diffusion property. Ultimately, the sensitivity of the RPA-CRISPR one-pot method

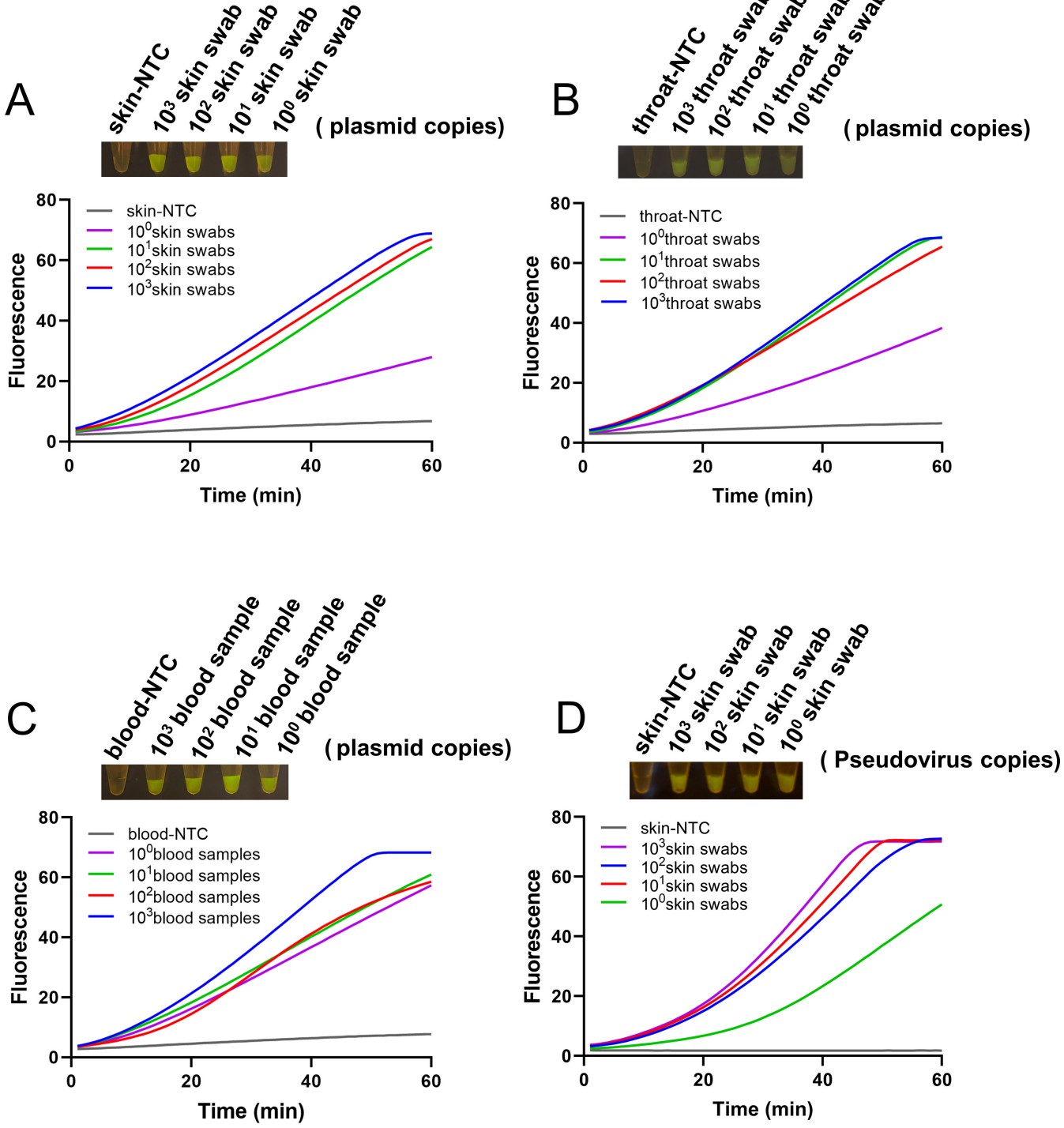

**FIG 7** Validation with simulated samples. Fluorescence curves and end-point fluorescence signals of reactions with simulated samples of skin swabs with the standard plasmid (A), throat swabs with the standard plasmid (B), blood samples with the standard plasmid (C), and skin swabs with the pseudovirus (D) are shown. The target amounts in the reactions are indicated. NTC represents the no-template controls.

for monkeypox detection reached a single copy per reaction. The optimization process of the sucrose system in this study indicates a wide redundancy in concentration and ratio, making it an easy-to-use principle for one-pot RPA-CRISPR methods.

With the multiphase aqueous system, the one-pot RPA-CRISPR method for the detection of monkeypox virus showed a sensitivity of a single copy per reaction, which

was higher than the existing RPA-CRISPR one-pot method by one magnitude (Table S1) (24). The detection method exhibited good specificity, effectively distinguishing monkeypox from other orthopoxviruses. Application of this one-pot RPA-CRISPR method has three advantages: (i) the high sensitivity reduces false-negative results and helps the early diagnosis of an infection; (ii) the amplification-cleavage system reduces false-positive results and gives better diagnostic accuracy; and (iii) the one-pot setting avoids exposing the amplified materials to the environment and reduces cross-contamination (19, 27, 28, 36). From the POCT perspective, the entire reaction was completed within 60 min at 37°C, and the detection results could be observed by the naked eye under ultraviolet or blue light. Although we used a qPCR machine to quantify the fluorescence signals in the method development, a portable fluorescence reader can do the quantification in real applications. For viral DNA extraction, a battery-powered mini-centrifuge is well enough. Thus, the method requires a set of pipettes, a mini-centrifuge, a low-power heat block, and a blue light source or a portable fluorescence reader. These supplies are easy to assemble, making the method highly suitable for home or limited healthcare settings.

In conclusion, this study utilized sucrose to increase the density of the RPA reaction and constructed a one-pot RPA-CRISPR method with a multiphase aqueous system, achieving rapid, convenient, and reliable detection of monkeypox virus. It assists in the prevention and control of monkeypox outbreaks and provides a good example of using a multiphase aqueous system to implement RPA-CRISPR one-pot methods.

## MATERIALS AND METHODS

### DNA sequences and plasmids

The standard plasmid representing a portion of the monkeypox virus F3L gene (GenBank no. ON568298/ON563414, 46,168–46,630 nt) in the background of pUC57 was purchased from Sangon Biotech (Shanghai) Co. Ltd., China. Two plasmids containing the SARS-CoV-2 N gene and ORF gene and the genomic DNA of *Pseudomonas aeruginosa* (*P. aeruginosa*, ATCC 9027) were preserved in the laboratory (38). Regions of the F3L homologous genes from other human-related orthopoxviruses, including the vaccinia virus (GenBank no. S64006.1, 170–630 nt of the E3L gene), the cowpox virus (GenBank no. NC_003663.2, 64,468–64,926 nt), the variola virus (GenBank no. DQ437592.1, 39,014–39,472 nt), and the buffalopox virus (GenBank no. MG599038.1, 51,078–51,536 nt), were synthesized in the background of pET28b(+) (General Biology Co. Ltd., Anhui, China). The sequence alignment of the homologous regions is shown in Fig. S1.

### RPA primers

The highly conserved F3L gene was selected as the detection target as described in the previous reports (39, 40). RPA primers were designed following the instructions of the RAA Nucleic Acid Amplification Reagent (Hangzhou ZC Bio-Sci & Tech Co. Ltd., Zhejiang, China) (Table 1; Fig. S1).

### PCR

RPA primers were initially screened by PCR. The PCR mixture (20 µL) contained 10 µL of the 2× Taq Master Mix (Vazyme Biotech Co. Ltd., Nanjing, China), 1 µL each of the forward and reverse primers (10 µM), and 5 µL of the template. The cycle setting was pre-denaturation at 95°C for 3 min followed by 30 cycles of 15 sec at 95°C, 15 sec at 55°C, and 60 sec at 72°C. The PCR amplicons were analyzed by agarose gel electrophoresis.

### RPA

The RPA reactions were conducted following the instructions of the RAA Nucleic Acid Amplification Reagent (Hangzhou ZC Bio-Sci & Tech Co. Ltd.). To a lyophilized enzyme

pellet, 25 µL of A buffer, 4 µL each of the forward and reverse primers (10 µM), and 8.5 µL of deionized water were added. The mixture was split into two halves for two reactions. Into each half of the mixture, 3 µL of the template DNA was added, and 1.25 µL of B buffer was added to start the reaction. The reaction (25 µL) was incubated at 37°C for 30 min. The amplicons were analyzed by agarose gel electrophoresis.

## Expression and purification of Cas12a

The coding sequence of Cas12a (LbCas12a) was obtained from a standard plasmid (Addgene plasmid no. 90096). The sequence was inserted into the multiple cloning site of the pET28b(+) vector by molecular cloning techniques, resulting in a recombinant open reading frame of an N-terminal His-tagged Cas12a for overexpression. The expression was induced with 0.5 mM IPTG for 8 h at 23°C in the *Escherichia coli* BL21(DE3)pLysS strain. The His-tagged recombinant Cas12a was purified by Ni affinity chromatography on a Ni-NTA 6FF column (Sangon Biotech Co. Ltd.) with a linear gradient of imidazole (0–500 mM) on an AKTA Prime Plus system (GE Healthcare Life Sciences, Boston, MA, USA). The purified Cas12a was analyzed by SDS-PAGE (Fig. S2) and stored at −20°C in 20 mM Tris-HCl, pH 8.0, 200 mM NaCl, 1 mM TCEP, and 50% (vol/vol) glycerol.

## crRNA preparation

The crRNAs were designed within the RPA amplicon sequences of the monkeypox virus F3L gene based on the available PAM motifs (TTTN) (Table 1; Fig. S1). For crRNA preparation, two reverse-complement ssDNA fragments that could form a dsDNA fragment containing the T7 promoter, the sequence for the crRNA skeleton, and the 21-bp sequence downstream of the corresponding PAM motif upon annealing were synthesized (General Biology Co. Ltd.). The annealed dsDNA served as the template for *in vitro* transcription with the T7 High Yield RNA Transcription Kit (Vazyme Biotech Co. Ltd.). The transcribed crRNA was purified by phenol-chloroform extraction, dissolved in 1,2-dierucoyl-sn-glycero-3-phosphocholine (DEPC) water, and quantified by a Qubit 4 fluorometer (Thermo Fisher Scientific Inc., Wilmington, DE, USA).

## Cas12a cleavage

The 20-µL Cas12a cleavage reaction mixture contained the desired amount of the DNA sample and 50 nM Cas12a, 50 nM crRNA, and 500 nM ssDNA-FQ (the fluorescence reporter with FAM and BHQ1 labels at the two ends) (Table 1) in 1× NEB Buffer 2.1. For real-time fluorescence detection, the reaction was conducted at 37°C for 15 min on the Roche LightCycler 480 II qPCR machine (Basel, Switzerland) with FAM signal reading at 1-min intervals. For end-point fluorescence detection, the signal was visualized under a blue light source after incubation at 37°C for 15 min. The images were taken by a smartphone camera.

## RPA-CRISPR one-pot reaction

The RPA premix and the CRISPR premix were prepared separately. The RPA premix was prepared by adding 4 µL each of the forward and reverse primers (10 µM) and 18 µL of A buffer to one lyophilized enzyme pellet (Hangzhou ZC Bio-Sci & Tech Co. Ltd.). The CRISPR premix was prepared by mixing 200 nM Cas12a, 2 µM ssDNA-FQ, and 200 nM crRNA in 1× NEB Buffer 2.1. The 25-µL RPA-CRISPR one-pot reaction was prepared by mixing 14.5 µL of the RPA premix, 6.25 µL of the CRISPR premix, and 3 µL of the template DNA and initiated by the addition of 1.25 µL of B buffer. The reaction was conducted at 37°C for 60 min on the Roche LightCycler 480 II qPCR machine (Basel, Switzerland) with FAM signal reading at 1-min intervals. The end-point signal was visualized under a blue light source, and the images were taken by a smartphone camera.

## RPA-CRISPR one-pot reaction with the sucrose-aided multiphase

The RPA premix and the CRISPR premix were prepared separately. For the RPA premix, 4 µL each of the forward and reverse primers (10 µM), 18 µL of A buffer, and 3.25 µL of sucrose (50%, vol/vol) were added to one lyophilized enzyme pellet (Hangzhou ZC Bio-Sci & Tech Co. Ltd.). The CRISPR premix was prepared by mixing 100 nM Cas12a, 1 µM ssDNA-FQ, and 100 nM crRNA in 1× NEB Buffer 2.1. To conduct the RPA-CRISPR one-pot reaction with the sucrose-aided multiphase, 18 µL of the RPA premix, 1 µL of the template, and 1 µL of B buffer were added to the bottom of the reaction tube and gently mixed. Then, 10 µL of the CRISPR premix was slowly added to the top of the above mixture, creating the sucrose-aided multiphase aqueous reaction system. The reaction was conducted at 37°C for 60 min on the Roche LightCycler 480 II qPCR machine (Basel, Switzerland) with FAM signal reading at 1-min intervals. The end-point signal was visualized under a blue light source, and the images were taken by a smartphone camera.

## Simulated samples

The skin swabs, throat swabs, and peripheral blood samples were collected from the consent-informed healthy persons. To prepare simulated skin swab and throat swab samples, the skin swabs or throat swabs were immersed in 2 mL of virus storage solution (Beyotime Biotechnology Co. Ltd., Shanghai, China), and then the solution was separated in 200-µL aliquots and mixed with the desired amounts of the monkeypox standard plasmid or the monkeypox pseudovirus (Fubio Biological Technology Co. Ltd., Shanghai, China). To prepare simulated blood samples, the collected blood samples were separated in 200-µL aliquots and mixed with desired amounts of the monkeypox standard plasmid or the monkeypox pseudovirus. DNA was extracted with the TIANamp Virus DNA/RNA Kit (Tiangen Biotech Co. Ltd., Beijing, China), and 1 µL was used for detection.

### ACKNOWLEDGMENTS

This work was supported by grants from the China Postdoctoral Science Foundation (No. 2022M721665), the Jiangsu Funding Program for Excellent Postdoctoral Talent of China (No. 2022ZB358), the Key Natural Science Research Project of the Jiangsu Higher Education Institutions of China (No. 20KJA416002), the Research Program of the "521 Project" of Lianyungang City of China (No. LYG06521202133), the Open-end Funds of Jiangsu Key Laboratory of Marine Bioresources and Environment (SH20221208), the "Blue Project" of Jiangsu Higher Education Institutions of China, and the Priority Academic Program Development of Jiangsu Higher Education Institutions of China.

### AUTHOR AFFILIATIONS

[1]Jiangsu Key Laboratory of Marine Biological Resources and Environment, Co-Innovation Center of Jiangsu Marine Bio-industry Technology, Jiangsu Key Laboratory of Marine Pharmaceutical Compound Screening, Jiangsu Ocean University, Lianyungang, China
[2]School of Food Science and Pharmaceutical Engineering, Nanjing Normal University, Nanjing, China

### AUTHOR ORCIDs

Pei Wang  http://orcid.org/0009-0008-5896-6963
Song Gao  http://orcid.org/0000-0002-0151-6493

## FUNDING

| Funder | Grant(s) | Author(s) |
|---|---|---|
| China Postdoctoral Science Foundation (China Postdoctoral Foundation Project) | 2022M721665 | Pei Wang |
| Jiangsu Funding Program for Excellent Postdoctoral Talent of China | 2022ZB358 | Pei Wang |

## AUTHOR CONTRIBUTIONS

Yue Wang, Conceptualization, Data curation, Formal analysis, Investigation, Methodology, Project administration, Software, Validation, Writing – original draft | Yixin Tang, Investigation, Methodology, Validation, Writing – original draft | Yukang Chen, Investigation, Methodology, Validation | Guangxi Yu, Investigation, Methodology | Xue Zhang, Formal analysis, Methodology | Lihong Yang, Methodology | Chenjie Zhao, Methodology, Software | Pei Wang, Conceptualization, Data curation, Formal analysis, Funding acquisition, Investigation, Methodology, Project administration, Writing – original draft, Writing – review and editing | Song Gao, Conceptualization, Data curation, Formal analysis, Funding acquisition, Project administration, Writing – original draft, Writing – review and editing

## DATA AVAILABILITY

The data presented in this study are included in the article; further inquiries can be directed to the corresponding authors.

## ADDITIONAL FILES

The following material is available online.

### Supplemental Material

**Supplemental material (Spectrum02267-23-s0001.pdf).** Fig. S1 and S2; Table S1.

### Open Peer Review

**PEER REVIEW HISTORY (review-history.pdf).** An accounting of the reviewer comments and feedback.

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
