## [Reviewer comments · Microbiology Spectrum]

Microbiology Spectrum

Ultrasensitive one-pot detection of monkeypox virus with RPA and CRISPR in a sucrose-aided multiphase aqueous system

Yue Wang, Yixin Tang, Yukang Chen, Guangxi Yu, Xue Zhang, Lihong Yang, Chenjie Zhao, Pei Wang, and Song Gao

Corresponding Author(s): Song Gao, Jiangsu Ocean University

Review Timeline:

Submission Date:	May 31, 2023
Editorial Decision:	September 4, 2023
Revision Received:	October 31, 2023
Accepted:	November 12, 2023

Editor: Frederick S. Kibenge

Reviewer(s): Disclosure of reviewer identity is with reference to reviewer comments included in decision letter(s). The following individuals involved in review of your submission have agreed to reveal their identity: Ruijie Deng (Reviewer #1); Wei Chen (Reviewer #2); Shuang Yang (Reviewer #3)

Transaction Report:

DOI: <https://doi.org/10.1128/spectrum.02267-23>

September 4, 2023

Dr. Song Gao
Jiangsu Ocean University
Jiangsu Province, Lianyungang City, Haizhou District, Tongguan south Road, No.108
Lianyungang
China

Re: Spectrum02267-23 (Ultrasensitive one-pot detection of monkeypox virus with RPA and CRISPR in a sucrose-aided multiphase aqueous system)

Dear Dr. Song Gao:

The reviewers recommend publishing your manuscript after the suggested revisions.

Link Not Available

Sincerely,

Frederick S. Kibenge

Journals Department
Reviewer comments:

Reviewer #1 (Comments for the Author):

The manuscript proposes a new method for the monkeypox virus detection. Due to the risk of monkeypox virus transmission and evolution, timely detection of monkeypox virus infection is essential to prevent monkeypox virus outbreaks. The authors developed a one-pot RPA-CRISPR assay for monkeypox based on a sucrose-aided multiphase aqueous system has been established with an ultra-high sensitivity of single copy, which is 10 times higher than the existing RPA-CRISPR one-pot method. The approach for monkeypox virus detection is valuable. However, there are still some issues should be addressed.

1. The manuscript established a sucrose-aided multiphase aqueous system for the one-pot detection of the monkeypox virus. By taking advantage of the density difference between sucrose and water, the separation of the RPA reaction and the CRISPR

- reaction within the same tube was realized. Can other reagents be used instead of sucrose? In addition, please discuss why sucrose does not interfere with protease response.
2. The authors optimized the concentrations of sucrose and the proportion of the RPA reaction system and CRISPR reaction system. But the concentrations of gRNA and Cas12a are important for CRISPR/system. Please explain the concentrations and basis of gRNA and Cas protein use.
 3. The manuscript compared the methods of recent studies on molecular detection of monkeypox, but some other Cas12a-based method should be discussed. For example, ACS Sens. 2021, 6, 9, 3295-3302; Nat. Biotechnol. 2020, 38, 870-874; J. Agric. Food Chem. 2021, 69, 35, 10321-10328; Nano Lett. 2021, 21, 19, 8393-8400.
 4. The P values from significance analysis is suggested to provide in sensitivity test.
 5. The expression and purification of Cas12a protein should be described with more details.
 6. More information about the selection of target RNA sequence is suggested to be provided.
 7. If possible, please test the clinical samples using the method.

Reviewer #2 (Comments for the Author):

1. The monkeypox is no longer a Public Health Emergency of International Concern as announced by WHO in May 2023. But it continues to pose a major public health challenge that requires a strong, proactive, and sustainable response. The authors have to update the information about the monkeypox epidemic and restate the significance of this study relating to the current situation.
2. There are several molecular detection methods using CRISPR in the literature. Also, there are RPA-based assays reported with high sensitivity. This study claimed an improvement on sensitivity with the one-pot CRISPR assay. The benefits of the higher sensitivity, the one-pot assay, and the use of CRISPR enzyme have to be further discussed to show the application advantage of this study.
3. The method established in this study is claimed to be used for POCT. However, DNA extraction has to be performed, and the authors described using a qPCR machine to read the signals. How would these be conducted in POCT? The authors have to clarify how to set up this assay in POCT, including the equipment and necessary training required.
4. The simulated samples used the plasmid with the monkeypox target sequence. This is an incomplete simulation because virus lysis and extraction of the genome are not covered. It is preferred to further validate the method with at least the pseudovirus.
5. Some of the experimental procedures and figures are not described with enough clarity. Suggest to improve the language of the Methods and Figure legends.
6. In Figure 6 A and B, the signal intensity of the sample with 10^2 copies is not consistent in the curve and the bar, please check the accuracy of the data.

Staff Comments:

Preparing Revision Guidelines

Please return the manuscript within 60 days; if you cannot complete the modification within this time period, please contact me. If

you do not wish to modify the manuscript and prefer to submit it to another journal, please notify me of your decision immediately so that the manuscript may be formally withdrawn from consideration by Microbiology Spectrum.

Spectrum02267-23

Ultrasensitive one-pot detection of monkeypox virus with RPA and CRISPR in a sucrose-aided multiphase aqueous system

Response to Reviewers: (responses in blue)

Reviewer #1 (Comments for the Author):

The manuscript proposes a new method for the monkeypox virus detection. Due to the risk of monkeypox virus transmission and evolution, timely detection of monkeypox virus infection is essential to prevent monkeypox virus outbreaks. The authors developed a one-pot RPA-CRISPR assay for monkeypox based on a sucrose-aided multiphase aqueous system has been established with an ultra-high sensitivity of single copy, which is 10 times higher than the existing RPA-CRISPR one-pot method. The approach for monkeypox virus detection is valuable. However, there are still some issues should be addressed.

We thank the positive comments from the reviewer.

1. The manuscript established a sucrose-aided multiphase aqueous system for the one-pot detection of the monkeypox virus. By taking advantage of the density difference between sucrose and water, the separation of the RPA reaction and the CRISPR reaction within the same tube was realized. Can other reagents be used instead of sucrose? In addition, please discuss why sucrose does not interfere with protease response. Glycerol can be used instead of sucrose. Sucrose and glycerol are traditional enzyme protectors, and studies have shown their enhancing effect on nucleic acid detections. This discussion has been added to the last paragraph of section “Principle of sucrose-aided multiphase one-pot RPA-CRISPR reaction” of RESULTS AND DISCUSSION.

2. The authors optimized the concentrations of sucrose and the proportion of the RPA reaction system and CRISPR reaction system. But the concentrations of gRNA and Cas12a are important for CRISPR/system. Please explain the concentrations and basis of gRNA and Cas protein use.

Regarding to the concentrations of Cas12a and crRNA, the literatures and the instruction of commercialized Cas12a (New England Biolabs) have recommended a molar ratio of ~1:1 in the range of 30-50 nM final. This study selected the molar ratio of 1:1 at 33 nM final for Cas12a and crRNA concentrations. This discussion has been added to the last paragraph of section “Construction of the one-pot RPA-CRISPR reaction in a sucrose-aided multiphase aqueous system” of RESULTS AND DISCUSSION.

3. The manuscript compared the methods of recent studies on molecular detection of monkeypox, but some other Cas12a-based method should be discussed. For example, ACS Sens. 2021, 6, 9, 3295-3302; Nat. Biotechnol. 2020, 38, 870-874; J. Agric. Food Chem. 2021, 69, 35, 10321-10328; Nano Lett. 2021, 21, 19, 8393-8400.

These literatures have been cited and discussed in the first paragraph of section “Advances on technology and application” of RESULTS AND DISCUSSION of the revised manuscript.

4. The P values from significance analysis is suggested to provide in sensitivity test.

The P values have been added to Figure 6B of the revised manuscript.

5. The expression and purification of Cas12a protein should be described with more details.

The result of expression and purification of Cas12a is shown in Figure S2. More details have been added to section “Expression and purification of Cas12a” of MATERIALS AND METHODS.

6. More information about the selection of target RNA sequence is suggested to be provided.

The detailed information about the selection of PAM motifs and the design of crRNAs are provided in Figure S1, and described with better clarity in section “crRNA preparation” of MATERIALS AND METHODS of the revised manuscript.

7. If possible, please test the clinical samples using the method.

Because of the strict control of the monkeypox virus clinical samples, we chose to use the monkeypox pseudovirus and validated the method with simulated samples. The results are presented in Figure 7 of the revised manuscript.

Reviewer #2 (Comments for the Author):

1. The monkeypox is no longer a Public Health Emergency of International Concern as announced by WHO in May 2023. But it continues to pose a major public health challenge that requires a strong, proactive, and sustainable response. The authors have to update the information about the monkeypox epidemic and restate the significance of this study relating to the current situation.

Changes have been made to the revised manuscript with the updated information in ABSTRACT, IMPORTANCE and the first paragraph of INTRODUCTION.

2. There are several molecular detection methods using CRISPR in the literature. Also, there are RPA-based assays reported with high sensitivity. This study claimed an improvement on sensitivity with the one-pot CRISPR assay. The benefits of the higher sensitivity, the one-pot assay, and the use of CRISPR enzyme have to be further discussed to show the application advantage of this study.

Application of this one-pot RPA-CRISPR method has three advantages: (1) the high sensitivity reduces false-negative results and helps early diagnosis of an infection; (2) the amplification-cleavage system reduces false-positive results and gives better diagnosis accuracy; (3) the one-pot setting avoids exposing the amplified materials to the environment and reduces cross-contaminations. A discussion has been added to the second last paragraph of section “Advances on technology and application” of RESULTS AND DISCUSSION of the revised manuscript.

3. The method established in this study is claimed to be used for POCT. However, DNA extraction has to be performed, and the authors described using a qPCR machine to read the signals. How would these be conducted in POCT? The authors have to clarify how to set up this assay in POCT, including the equipment and necessary training required.

Although we used a qPCR machine to quantify the fluorescence signals in the method development, a portable fluorescence reader can do the quantification in real applications. For viral DNA extraction, a battery-powered mini-centrifuge is well enough. Thus, the method requires a set of pipettes, a mini-centrifuge, a low-power heat block, and a blue light or a portable fluorescence reader. These supplies are easy to assemble, making the method highly suitable for POCT. A discussion has been added to the

second last paragraph of section “Advances on technology and application” of RESULTS AND DISCUSSION of the revised manuscript.

4. The simulated samples used the plasmid with the monkeypox target sequence. This is an incomplete simulation because virus lysis and extraction of the genome are not covered. It is preferred to further validate the method with at least the pseudovirus.

Because of the strict control of the monkeypox virus clinical samples, we chose to use the monkeypox pseudovirus and validated the method with simulated samples. The results are presented in Figure 7 of the revised manuscript.

5. Some of the experimental procedures and figures are not described with enough clarity. Suggest to improve the language of the Methods and Figure legends.

The language of the Methods and Figure legends has been modified in the revised manuscript. We believe the language is now satisfactory, presenting information with good clarity.

6. In Figure 6 A and B, the signal intensity of the sample with 10^2 copies is not consistent in the curve and the bar, please check the accuracy of the data.

We had made a mistake in the bar chart of Figure 6B. It is now corrected in the revised manuscript.

Re: Spectrum02267-23R1 (Ultrasensitive one-pot detection of monkeypox virus with RPA and CRISPR in a sucrose-aided multiphase aqueous system)

Dear Dr. Song Gao:

The Reviewers' comments have been adequately addressed.

Your manuscript has been accepted, and I am forwarding it to the ASM production staff for publication. Your paper will first be checked to make sure all elements meet the technical requirements. ASM staff will contact you if anything needs to be revised before copyediting and production can begin. Otherwise, you will be notified when your proofs are ready to be viewed.

Sincerely,
Frederick S. Kibenge
Editor
Microbiology Spectrum

Reviewer #1 (Comments for the Author):

Authors have addressed the issues.